# Biofilm Inhibition and Antimicrobial Properties of Silver-Ion-Exchanged Zeolite A against *Vibrio* spp. Marine Pathogens

Zarina Amin [1], Nur Ariffah Waly [2] and Sazmal Effendi Arshad [2,*]

1   Biotechnology Research Institute, Universiti Malaysia Sabah, Kota Kinabalu 88400, Malaysia; zamin@ums.edu.my
2   Faculty of Science & Natural Resources, Universiti Malaysia Sabah, Kota Kinabalu 88400, Malaysia; arifahwaly@gmail.com
*   Correspondence: sazmal@ums.edu.my; Tel.: +60-178180618

**Featured Application: This study highlights a potential application of ion exchanged zeolite A against marine microbial pathogens and their biofilms.**

**Abstract:** A challenging problem in the aquaculture industry is bacterial disease outbreaks, which result in the global reduction in fish supply and foodborne outbreaks. Biofilms in marine pathogens protect against antimicrobial treatment and host immune defense. Zeolites are minerals of volcanic origin made from crystalline aluminosilicates, which are useful in agriculture and in environmental management. In this study, silver-ion-exchanged zeolite A of four concentrations; 0.25 M (AgZ1), 0.50 M (AgZ2), 1.00 M (AgZ3) and 1.50 M (AgZ4) were investigated for biofilm inhibition and antimicrobial properties against two predominant marine pathogens, *V. campbelli* and *V. parahemolyticus*, by employing the minimum inhibitory concentration (MIC) and crystal violet biofilm quantification assays as well as scanning electron microscopy. In the first instance, all zeolite samples AgZ1–AgZ4 showed antimicrobial activity for both pathogens. For *V. campbellii*, AgZ4 exhibited the highest MIC at 125.00 µg/mL, while for *V. parahaemolyticus*, the highest MIC was observed for AgZ3 at 62.50 µg/mL. At sublethal concentration, biofilm inhibition of *V. campbelli* and *V. parahemolyticus* by AgZ4 was observed at 60.2 and 77.3% inhibition, respectively. Scanning electron microscopy exhibited profound structural alteration of the biofilm matrix by AgZ4. This is the first known study that highlights the potential application of ion-exchanged zeolite A against marine pathogens and their biofilms.

**Keywords:** biofilms; zeolite A; *Vibrio* spp.; antimicrobials

## 1. Introduction

Aquaculture farming has been the fastest growing food producing sector in the last few decades and an important industry in many developing countries. However, the industry currently faces a threatening challenge due to the bacterial disease outbreaks resulting in high mortality rates in the aquaculture population [1,2]. This is in part due to extensive use of antibiotics in fish farms leading to antimicrobial resistance in fish pathogens [3–5]. Vibriosis is an important bacterial disease in wild and farmed marine fishes, which results in severe economic loss of more than USD 1 billion [6]. In addition, bacteria from seafood sources have been associated with foodborne outbreaks [7].

The formation of biofilms by marine pathogens further amplifies problems faced in the aquatic systems and the aquaculture industry. Biofilms are self-assembled communities of bacteria embedded in a self-developed extracellular matrix (ECM) and are adherent to abiotic or abiotic surfaces. The ECM contains exopolysaccharides, proteins and extracellular DNA (eDNA) [8–10]. Biofilm formation is induced by genetic factors and influenced by environmental conditions including pH, temperature and the availability of nutrients [9].

The formation of ECM in biofilms presents a physical barrier that allows the bacteria to protect themselves from external disturbances. This further leads to the improvement of their resilience to the external environmental conditions and enhance their virulence to cause disease. Examples of enhanced resilience include resistance against antimicrobial agents, disinfectants and host defense mechanisms, making them more difficult and expensive to treat [11,12].

The extensive use of antibiotics in marine systems coupled with the enhanced resilience of marine pathogens as biofilms leads to a "double whammy" scenario of antimicrobial resistance in the aquaculture industry, resulting in huge economic losses due to high mortality rates of fish and seafood as important commodities in countries with high economic dependency on seafood farming such as South East Asia and Japan. The prevalent challenge has necessitated efforts on new antibacterial materials that can effectively inhibit growth and resistance of aquatic pathogens while minimizing negative impacts on human and animal health and the environment.

*V. parahemolyticus* is a prevalent food-poisoning bacterium associated with seafood consumption, typically causing self-limiting gastroenteritis and commonly found in temperate and tropical marine and coastal waters globally [13,14]. Vibriosis is a systemic bacterial infection in farmed and wild marine fishes, which is considered to be a profoundly significant problem due to intensive economic losses in aquaculture industry worldwide [15]. *V. campbelli* is as an emerging marine pathogen recently associated with diseased farm shrimps [16].

Several studies have reported on the development of bacterial biofilms in aquatic environments, particularly challenges faced due to the persistence of these biofilms as important sources of infection and disease in the aquaculture industry. The diseases result in huge economic losses due to high mortality rates of seafood as important commodities [17–19]. Several strategies have been evaluated to combat biofilm development in aquaculture systems, and these include seawater ozonation [20], use of probiotics [21] and sanitizers [22]. In a recent minireview by Artunes et al., the strategy of quorum sensing (QS) inhibition was discussed. QS refers to chemical signaling molecules that control biofilm formation in bacteria [23]. Recently, several studies on antibacterial agents for the aquaculture systems have also been explored. These include non-chemotherapeutic methods such as natural therapeutics from plants and immunostimulants [24–26] and by alternative inorganic materials [27]. Additionally, studies on the application of zeolites as an antimicrobial agent have started to gain traction particularly in the field of aquaculture.

Zeolite is a microporous, crystalline aluminosilicate with a framework that is made up of $[SiO_4]^{4-}$ and $[AlO_4]^{5-}$ tetrahedra. The tetrahedra are joined together by sharing their corner oxygen atom [28]. In this framework, positively charged silicon ion was balanced by oxygen ion. While the negative charged alumina remains unbalanced resulting in a negative charge of the total structure of zeolite, and it is balanced by extra framework cations [29]. These cations can be exchanged with any other positive ions in order to modify their properties to match with the desired application.

Although there has been extensive research on the antibacterial activity of ion-loaded zeolite against common bacteria including *Escherichia coli, Bacillus subtilis, Staphylococcus aureus, Klebsiella pneumoniae, Pseudomonas aeruginosa* and wastewater biofilm bacteria [30–33], to our knowledge, however, there is still relatively a small number of studies of the effectivity of ion-loaded zeolite A against marine pathogens, in particular *Vibrio campbellii* and *Vibrio parahaemolyticus*.

This study aimed to investigate antimicrobial and antibiofilm applications of silver-ion-exchanged zeolite A against marine bacterial pathogens *V. parahaemolyticus* and *V. campbelli*.

## 2. Materials and Methods

### 2.1. Liquid Ion Exchange of Zeolite A

The initial synthesis of zeolite A from bentonite clay comprised the activation of precursor bentonite clay by thermochemical treatments in HCl, addition of alkaline activators, ageing and crystallization processes as well as characterization by X-ray diffraction (XRD) and scanning electron microscopy (SEM) analysis (data published elsewhere).

For liquid ion exchange, the integration of silver (Ag) ion into zeolite A was performed by using the liquid ion exchange method as described by Demirci et al. [31], with slight modifications. Briefly, 1 g zeolite A was added into 10 mL of four $AgNO_3$ concentrations: 0.25, 0.50, 1.00 and 1.50 M (denoted AgZ1, AgZ2, AgZ3 and AgZ4, respectively) and mixed at 150 rpm for 3 days. The obtained zeolite A were then vacuum filtered, washed with deionized water and dried at 80 °C overnight.

For the metal ion concentration analysis, a Perkin-Elmer Optima 5300DV (Waltham, MA, USA) axial viewing ICP-OES was used. The analytical wavelengths of elements included were: Na 589.592, Ag 328.068 and Cu 327.393 nm. The software used was WinLab32 for ICP. Briefly, 200 mg of AgZ1, AgZ2, AgZ3 and AgZ4 was added into separate tubes containing 14 mL of acid mixture, aqua regia (40% $HNO_3$ + 60% HCl) and left overnight at room temperature. Four milliliters of aqua regia solution was then added into the tubes and mixed for 30 min at 80 °C before being filtered and diluted prior the ICP-OES analysis.

### 2.2. Minimum Inhibition Concentration (MIC) Assay of Ag-Exchanged Zeolite A

The minimal inhibitory concentration (MIC) assay of Ag-exchanged zeolite A against *V. parahaemolyticus* and *V. campbellii* was carried out as described [33]. Briefly, frozen stocks of *V. campbellii* and *V. parahaemolyticus* were grown on nutrient agar (NA) supplemented with 2% of NaCl and incubated for 16 h at 37 °C. Cultures were then inoculated into tryptone soy broth (TSB), and absorbances were adjusted accordingly to an initial starting $OD_{600}$ of 0.05. As seen in Table 1, samples AgZ1, AgZ2, AgZ3 and AgZ4 of respective AgZ concentrations were then added. The assay for each sample was tested with two technical replicates.

**Table 1.** Silver ion content (mg/g) of zeolite of samples.

| Metal Ion/Concentration | 0.25 M (1) | 0.50 M (2) | 1.00 M (3) | 1.50 M (4) |
|---|---|---|---|---|
| Silver (AgZ) | 80.949 ± 0.389 | 160.041 ± 1.328 | 177.481 ± 1.483 | 190.511 ± 1.989 |

For each bacterial species, doubling dilutions up to 1/512 of AgZ1, AgZ2, AgZ3 and AgZ4 were firstly carried out. Five hundred microliters of liquid inoculum ($OD_{600}$ adjusted to 0.05) was then added and incubated in a shaking 37 °C incubator for 16 h. One hundred microliters of the culture was then plated and spread on to nutrient agar (NA) plates supplemented with 2% NaCl and further incubated for 16 h at 37 °C with appropriate controls. The minimum inhibitory concentration (MIC) of AgZ1, AgZ2, AgZ3 and AgZ4 was determined by the lowest concentration of samples that did not exhibit bacterial growth.

### 2.3. Quantitation of Bacterial Biofilm Growth by Crystal Violet Assay

The biofilm growth of AgZ-treated biofilm were quantitated by the crystal violet assay using the methods described by O'Toole [34] with modifications. Briefly, cultures of *V. campbellii* and *V. parahaemolyticus* were incubated for 16 h at 37 °C in tryptone soy broth (TSB) supplemented with 2% NaCl.

For the biofilm assay, the sublethal concentration obtained from the MIC assay from was selected as the starting assay concentration. Briefly, serial two-fold dilutions of the sample were performed in a 96-well microtiter plate (corning) containing tryptone soy

broth supplemented with 2% NaCl. Fifty microliters of the overnight cultures (with $OD_{600}$ adjusted to 0.05) were then inoculated into each well, and it was then incubated at 37 °C for 72 h. Following incubation, wells were washed three times with distilled $H_2O$, desiccated at 50 °C, stained with 1% crystal violet for 30 min and added with 95% ethanol. The absorbance of solubilized dye was then determined at 570 nm (Shimadzu, Spectramax, Kyoto, Japan).

### 2.4. Scanning Electron Microscopy (SEM) of V. campbellii and V. parahaemolyticus Biofilms

The cell morphologies of *V. campbellii* and *V. parahaemolyticus* biofilms grown in TSB media and in TSB with AgZ4 were analyzed by SEM. Briefly, 5 mL overnight liquid cultures of *V. campbellii* and *V. parahaemolyticus* were cell fixated according to protocol by Gomes and Mergulhão [35] with modifications. The biofilm samples were then fixed in 5% glutaraldehyde prepared in 0.1 M PBS pH 7.2 at 4 °C for 12 h. Following the fixation process, the samples were dehydrated by introduction into a series of ethanol solution of varying concentration gradients (35, 50, 75, 95 and 2 × 100%). The dehydrated samples were then immersed in HMDS for 10 min. Upon dehydration, the samples were dried overnight and then sputter-coated with platinum before being analyzed by SEM (Hitachi SEM, S 3400N, Tokyo, Japan).

### 3. Results

The incorporation of silver ion (Ag) into the zeolite A framework (AgZ) was determined by introducing the zeolite sample to four different concentrations of Ag solution AgZ1, AgZ2, AgZ3 and AgZ4. During this process, the potassium and sodium ions that exist in the zeolite framework were exchanged by metal ions. ICP-OES analysis of AgZ1, AgZ2, AgZ3 and AgZ4 is as shown in Table 1.

As seen in Table 1, the highest silver ion content was observed for AgZ4 at 190.511 ± 1.989 mg/g, followed by the lowest concentration for AgZ1 at 80.949 ± 0.389 mg/g.

### 3.1. Minimum Inhibitory Concentration (MIC) of V. campbellii and V. parahaemolyticus in AgZ

As seen in Table 2, both *V. campbellii* and *V. parahaemolyticus* showed susceptibility to all four AgZ1-AgZ4 concentrations. For *V. campbellii*, the highest MIC was observed in AgZ4 at 0.1250 mg/mL. For *V. parahaemolyticus*, the highest MIC was observed in AgZ3 at 0.0625 mg/mL. The lowest MIC for both *V. campbellii* and *V. parahaemolyticus* was observed in AgZ1 at 2.0 and 0.5 mg/mL, respectively.

**Table 2.** Minimum inhibitory concentration (MIC) of silver-ion-loaded zeolite A against *V. campbellii* and *V. parahaemolyticus*.

| | MIC Value of Four Different Silver-Ion-Loaded Zeolite A (mg/mL) | | | |
|---|---|---|---|---|
| Bacterial Pathogens | AgZ1 | AgZ2 | AgZ3 | AgZ4 |
| *V. campbellii* | 2.0000 | 0.5000 | 0.2500 | 0.1250 |
| *V. parahaemolyticus* | 0.5000 | 0.1250 | 0.0625 | 0.0625 |

### 3.2. Quantitation of Bacterial Biofilm Density by Crystal Violet Assay

The AgZ4 MIC concentration for both pathogens were further selected for biofilm inhibition crystal violet assay. Figure 1 demonstrated biofilm inhibition percentages of ion-exchanged zeolite in comparison with the untreated control. As shown in Figure 1, at 570 nm absorbance, biofilm formation of AgZ4 against *V. campbellii* and *V. parahaemolyticus* exhibited up to 60.2 and 77.3% inhibition, respectively.

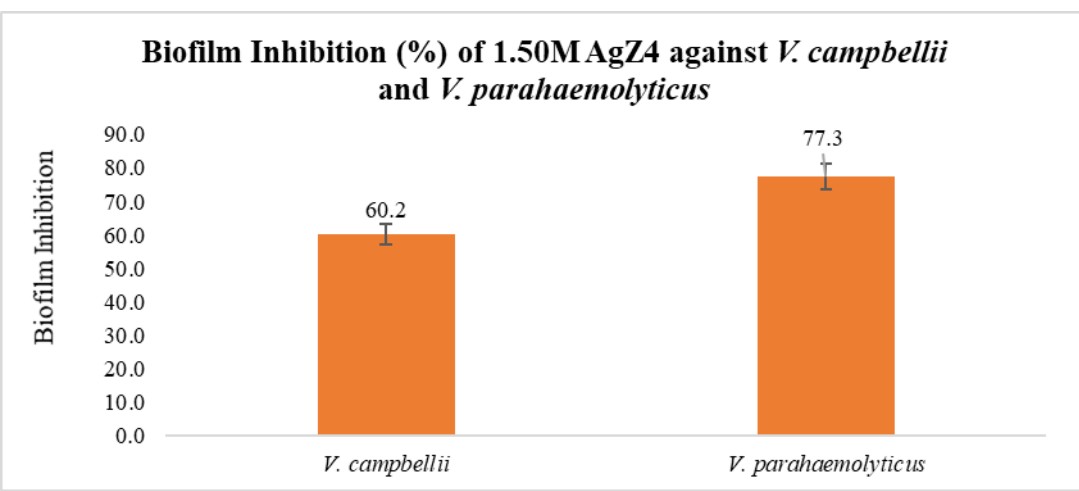

**Figure 1.** Biofilm inhibition of *V. campbellii* and *V. parahaemolyticus* by 1.50 M AgZ4 assessed by CV staining. Biofilm inhibition of *V. campbellii* and *V. parahaemolyticus* isolates after 24 h growth with 1.50 M AgZ4 was assayed by CV staining (A570).

Scanning Electron Microscopy of *V. campbellii* and *V. parahaemolyticus* in AgZ4, Figures 2a and 3a represent the SEM images of untreated samples of *V. campbellii* and *V. parahaemolyticus*, respectively, while Figures 2b and 3b represent the SEM images of *V. campbellii* and *V. parahaemolyticus* after growth in media culture treated with 1.50 M AgZ4.

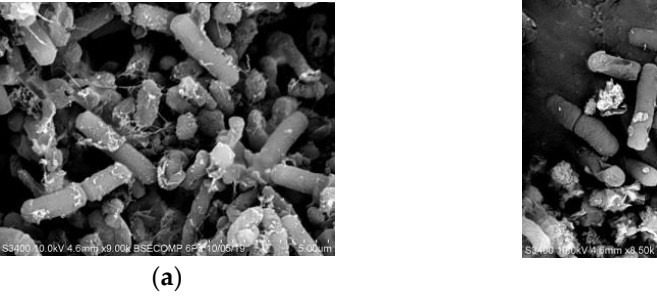

(**a**)　　　　　　　　　　　　　　(**b**)

**Figure 2.** SEM analysis showing (**a**) untreated *V. campbellii* biofilm bacteria embedded in extracellular matrix biofilm and (**b**) *V. campbellii* treated with AgZ4 where profound loss of biofilm matrix is seen.

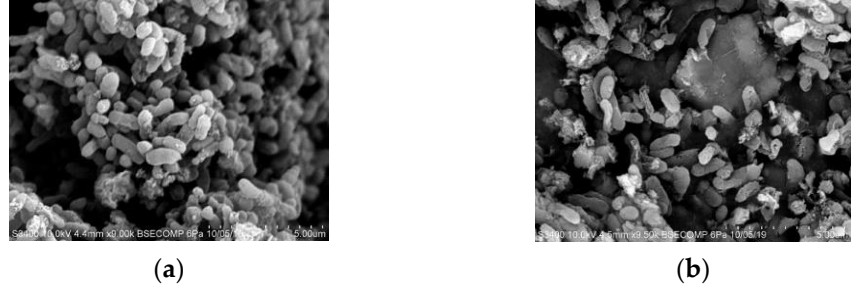

(**a**)　　　　　　　　　　　　　　(**b**)

**Figure 3.** SEM analysis showing (**a**) untreated *V. parahaemolyticus* biofilm bacteria embedded in extracellular matrix biofilm and (**b**) *V. parahaemolyticus* treated with AgZ4 where profound loss of biofilm matrix is seen.

Figure 2a shows clustering and aggregation of *V. campbellii* covered with a network of extracellular matrix (ECM) as a typical representation of intact biofilms. In contrast, Figure 2b shows markedly reduced ECM and aggregation of bacteria.

Similarly, Figure 3a below shows clustering and aggregation of untreated *V. parahaemolyticus* covered with a fine network of extracellular matrix (ECM). In contrast, as seen

in Figure 3b, *V. parahaemolyticus* exposure to AgZ4 showed a markedly reduced ECM and cell clustering.

## 4. Discussion

This study attempted to evaluate the antimicrobial and antibiofilm activities of silver-ion-exchanged zeolite (AgZ) against two marine bacterial pathogens, *V. campbellii and V. parahaemolyticus*. The first stage of this study involved the ICP-OES evaluation of Ag ion incorporation of zeolite synthesized from local bentonite clay (data published elsewhere). The initial ICP-OES analysis of AgZ1, AgZ2, AgZ3 and AgZ4 revealed the successful incorporation of Ag ion into the zeolite samples to be typically in direct correlation with the increase in Ag ion concentration introduced. The highest Ag ion concentration was observed in sample AgZ4 (1.50 M) at 190.511 ± 1.989 mg/g. At 0.50 M, the Ag ion concentration of AgZ2 was double that of AgZ1 at 160.041 ± 1.328 mg/g compared to 80.949 ± 0.389 mg/g. Nonetheless, doubled concentration increases observed for AgZ2 were not indicated in AgZ3 and AgZ4. A possible explanation for this is saturation of available sites in AgZ3 and AgZ4 for the ion exchange to occur.

In general, minimum inhibitory concentration (MIC) evaluation against *V. campbellii* and *V. parahaemolyticus* firstly revealed that the incorporation of Ag ions into the zeolite samples AgZ1, AgZ2, AgZ3 and AgZ4 was successful in inhibiting bacterial growth. Silver (Ag) has long been established in various studies as an effective antibacterial agent. Several studies have also demonstrated the efficiency of Ag-exchanged zeolite against many microbial pathogens [31–33] and concur with the findings of this study. Additionally, a typically inverse relationship between sample concentration and MIC for AgZ1, AgZ2, AgZ3 and AgZ4 was also observed. As seen in Table 2 a lower MIC, which signifies stronger antibacterial activity, was indicated for each bacterial type as the concentration of metal ion in the zeolite increased. Therefore, the higher the Ag ion loading in the zeolite, the lower the concentration of ion-loaded zeolite A needed to inhibit the growth of the bacteria. There have been many studies that have investigated the mechanism of microbial killing by metal ion. A possible mechanism involves the ability of the metal ion to attach to the bacteria membrane through electrostatic interaction and drastically alter the integrity of the bacterial membrane. Consequently, it promotes the formation of reactive oxygen species, (ROS) which will induce the oxidative stress to the bacteria cell resulting in the oxidation of cellular component, DNA damage, mitochondria damage and disruption of the cell membrane, which lead to the death of the bacteria [36–39].

MIC studies also revealed that between the two bacterial species, *V. parahaemolyticus* indicated higher susceptibility against Ag ion compared to *V. campbellii*, as lower MICs were observed across AgZ1–AgZ4. Under antibiotic pressure, bacterial phenotypes such as susceptibility, resistance, tolerance and persistence differ from one bacteria to the other. In a review by Li et al. [40], the efficacy of antimicrobials are influenced by many factors including bacterial status, host factors and antimicrobial concentrations.

A typical feature of bacterial biofilms is the extracellular matrix, which provides protection and structure to the cell population within it. The CV assay is a useful tool for rapid and simple assessment of biofilm formation differences between bacteria, as it stains the extracellular matrix as well as the aggregated bacterial cells [41]. The CV biofilm assays of *V. campbelli* and *V. parahaemolyticus* isolates grown in AgZ4 overnight showed that biofilm growth was effectively inhibited in the presence of AgZ4. However, *V. parahaemolyticus* biofilms were indicated to be more susceptible against AgZ4 with a percentage inhibition of 77.3% compared to *V. campbelli* at 60.2%. This concurred with the MIC assays, which also showed a higher susceptibility of *V. parahaemolyticus* when compared against *V. campbelli* and demonstrated significant attenuation of biofilm formation against *V. campbellii* and *V. parahaemolyticus*.

The ability of Ag ion-exchanged zeolite AgZ4 to disrupt biofilm development for *V. campbellii* and *V. parahaemolyticus* was further supported by SEM analysis. As seen in Figures 2 and 3, the exposure of AgZ4 to both *V. campbellii* and *V. parahaemolyticus* displayed

significant structural alteration of biofilm phenotypes when compared to bacterial isolates grown in the absence of AgZ4, including profound loss of the biofilm extracellular matrix (ECM) as well as markedly reduced cell aggregation. While SEM analysis of untreated isolates showed tight aggregation of cells held together by ECM, isolates exposed to AgZ4 showed higher numbers of singular isolates with lessened clustering. Breakages on the extracellular matrices of biofilms will result in increased susceptibility of bacteria against antibacterial agents and chemicals [42]. Despite much literature on antibiofilm activities of bacteria by zeolite, the mechanism of toxicity of AgZ against biofilms of *V. campbellii* and *V. parahaemolyticus* is still poorly understood. Therefore, future studies on the probability of modification of gene expression in the *Vibrio* polysaccharide (VPS) and matrix protein biosynthesis are recommended to further inform on genes or protein that are significantly affected by metal-loaded zeolites.

## 5. Conclusions

The study findings strongly indicate antimicrobial and antibiofilm characteristics of the silver-ion-exchanged zeolite A against the bacterial pathogens, with the highest MIC levels observed for AgZ4 (1.50 M) for *V. campbelli* and AgZ3 (1.00 M) for *V. parahaemolyticus*. Scanning electron microscopy exhibited profound breakages in the biofilm structures of both marine pathogens when grown in media added with 1.50 M silver-ion-exchanged zeolite A (AgZ4). Taken together, the results of this study strongly indicate the strong antibacterial and antibiofilm potentials of Ag-ion-exchanged zeolite A, which can be applied in the aquaculture industry to combat against infectious pathogens, in particular *V. campbellii* and *V. parahaemolyticus*.

**Author Contributions:** Formal analysis, Z.A. and N.A.W.; Supervision, S.E.A.; Writing—original draft, Z.A.; Writing—review & editing, Z.A. All authors have read and agreed to the published version of the manuscript.

**Funding:** This research was funded by Universiti Malaysia Sabah, grant number GUG0111-1/2017. The APC was funded by Universiti Malaysia Sabah.

**Informed Consent Statement:** Not applicable.

**Data Availability Statement:** Not applicable.

**Conflicts of Interest:** The authors declare no conflict of interest.

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
