# Peer review of "Biofilm Inhibition and Antimicrobial Properties of Silver-Ion-Exchanged Zeolite A against Vibrio spp Marine Pathogens"

_applsci, doi:10.3390/app11125496_

Round 1

Reviewer 1 Report

The paper of Amin et al is focussed on the antimicrobial effect of silver ion exchange zeolite A. The originality of paper is low, but the applicability in aquaculture of this research is obvious.

The authors have to present the ICP OES equipment, the sample introduction system, the operational and measuring systems, the software, etc.

The lines 210-215 have to be removed because the same information is presented at the end of introduction.

Author Response

Comments and Suggestions for Authors (Reviewer 1)

1.The paper of Amin et al is focussed on the antimicrobial effect of silver ion exchange zeolite A. The originality of paper is low, but the applicability in aquaculture of this research is obvious.

Authors’ response : Comment noted.

2.The authors have to present the ICP OES equipment, the sample introduction system, the operational and measuring systems, the software, etc.

Authors’ response : The statement “For the metal ion concentration analysis, a Perkin-Elmer Optima 5300DV axial viewing ICP-OES was used. The analytical wavelengths of elements included were: Na 589.592 nm, Ag 328.068 nm and Cu 327.393 nm. The software used was WinLab32 for ICP.” was added into the appropriate methodology section. Please see lines 112-114 in manuscript.

3.The lines 210-215 have to be removed because the same information is presented at the end of introduction.

Authors’ response : Lines 210-215 removed. Please see manuscript.

Reviewer 2 Report

Comment 1: The author should include one more experiment regarding the toxicity of ion-exchanged Zeolite. 

The concentration AgZ (0.25M - 1.5M) is safe for fish, shellfish, aquatic plants (seaweed) ??

Comment 2: References are missing in the Materials and method section.

Author Response

Comments and Suggestions for Authors (Reviewer 2)

1.Comment 1: The author should include one more experiment regarding the toxicity of ion-exchanged Zeolite. 

Authors’ response : This is study involves a preliminary evaluation antibacterial and antibiofilm activities of Ag-exchanged zeolite A against two marine pathogens . This preliminary stage study did not involve any fish challenge with the Ag-exchanged zeolite. The second stage of this study which be carried out subsequently, will involve exposure of silver-exchanged zeolites into the aquatic environments of fish and seafood by feeding . The need to address toxicity will be addressed in the second consecutive stage of study when the specific aquatic organisms are used.

2.The concentration AgZ (0.25M - 1.5M) is safe for fish, shellfish, aquatic plants (seaweed) ??

Authors’ response : 1) A study by our group that has been accepted for publication in Songklanakarin Journal of Science and Technology titled “Effects of sodium bentonite clay as a feed additive on the growth and haematology parameters of hybrid grouper Epinephelus fuscoguttatus x Epinephelus lanceolatus” strongly indicated that the use of 1.5% sodium bentonite clay (zeolite) did not impair the growth of the hybrid grouper, Epinephelus fuscoguttatus x Epinephelus lanceolatus (Article in Press: SJST-2019-0312.R1 https://rdo.psu.ac.th/sjstweb/ArticleInPress.php )

2) Another study by our group (accepted in International Journal of Fisheries and Aquatic Studies
http://www.fisheriesjournal.com/) titled “Nutritional and Immunogenetic Profiling Reveal the Effects of Dietary Zeolite on Asian Seabass Lates calcarifer” indicated that dietary zeolite supplementation boosted overall Body Weight Gain (BWG) with fish fed with Z4 (4% zeolite); showing the highest BWG. Feed Conversion Ratios (FCR) and Protein Efficiency Ratio (PER) were also significantly improved (P < 0.05) by the supplementation of 2, 4 and 8% zeolite. Importantly, immune response genes analysis exhibited no significant changes in the immune and stress responses of fish at both 4% and 8% zeolite supplementation, strongly indicating the absence of immune impairment to L. calcarifer.

3.Comment 2: References are missing in the Materials and method section.

Authors’ response : References are added for each methodology as appropriate. Please see References are 31(line 107), 33 (line 121), 34 (line 136), 35 (line 152)

Please refer to manuscript.

Reviewer 3 Report

In this manuscript, the Authors describe the potential application of ion-exchanged Zeolite-A against two Vibrio spp marine pathogens. Specifically, the Authors focus their study on antimicrobial and antibiofilm activity of silver ion-exchanged zeolite A against V. campbelli and V. parahemolyticus.

Although the results appear very interesting, the work is missing in some experiments and should be rewritten in all sections.

Specifically, the manuscript needs of the following implementations:

INTRODUCTION

The introduction should be improved, motivating the choice of the two pathogenic strains and adding the potential applications of zeolite in aquaculture. At this purpose, I would suggest to move the discussion from line 201 to 210  (V. parahemolyticus is a prevalent food-poisoning bacterium associated with seafood consumption… applications in aquaculture especially in the purification of hatchery tanks (25)” in the Introduction section.

  • Similarly, the sentences from line 54 to line 59 (“Study findings strongly indicated antimicrobial and antibiofilm characteristics of the …… zeolite A against marine pathogens and their biofilms.”) should be moved in Conclusion section.

MATERIAL AND METHODS

  • Line 83-84: Authors should indicate the incubation time, considering that the culture for the MIC assay must be in the semi-exponential phase
  • Table 1: should be removed as it does not add any additional information
  • Line 93: Authors should be changed “doubling dilutions” in “serially two-fold dilution
  • Line 94: Mic assay is generally performed using 105 bacteria/mL (although higher concentration inoculums are used in many studies). Therefore, the Authors should indicate bacterial concentration corresponding to OD600 = 0.05.
  • Line 101: Are the authors evaluating AgZ for inhibition of biofilm formation (and not for degradation of a formed biofilm)? If the answer is yes, they should change "The biofilm density of AgZ treated biofilm" to "Biofilm inbition activity by AgZ"

RESULTS

  • About MIC assay. The data shown in table 2 are partially in contrast with those of table 3. This result could depend on a different release over time of the Ag ions from the various AgZ complexes. Therefore, I suggest adding an Ag-release test over time (up to 16 hours) to validate the results obtained.
  • About Biofilm inhibition. The Authors should also add the values of the other AgZ (AgZ1, AgZ2 and AgZ3). From the MIC data, I would expect similar data between AgZ3-AgZ4 for parahemolyticus.
  • About Figure 1. The Authors should indicate the number of replicates from which the average value was obtained. In addition, if the replicas are at least 5, a test-anova should be done to assess whether the differences between the two strains are significant.
  • I would also suggest doing other tests, such as live/dead or XTT, to increase biofilm inhibition results in terms of live and dead cell ratio and toxicity, respectively.
  • About Scanning Electron Microscopy (SEM). Similar to the Biofilm assay, I would suggest to increase the data with the other AgZ (AgZ1, AgZ2 and AgZ3).

Author Response

Comments and Suggestions for Authors (Reviewer 3)

  1. In this manuscript, the Authors describe the potential application of ion-exchanged Zeolite-A against two Vibrio spp marine pathogens. Specifically, the Authors focus their study on antimicrobial and antibiofilm activity of silver ion-exchanged zeolite A against V. campbelli andV. parahemolyticus.

Although the results appear very interesting, the work is missing in some experiments and should be rewritten in all sections.

Specifically, the manuscript needs of the following implementations:

  1. INTRODUCTION

The introduction should be improved, motivating the choice of the two pathogenic strains and adding the potential applications of zeolite in aquaculture. At this purpose, I would suggest to move the discussion from line 201 to 210  (V. parahemolyticus is a prevalent food-poisoning bacterium associated with seafood consumption… applications in aquaculture especially in the purification of hatchery tanks (25)” in the Introduction section.

Authors’ response : Line 201 to 210 in Discussion has been moved to the Introduction section as suggested. Please refer to lines 63-69 of the manuscript.

Similarly, the sentences from line 54 to line 59 (“Study findings strongly indicated antimicrobial and antibiofilm characteristics of the …… zeolite A against marine pathogens and their biofilms.”) should be moved in Conclusion section.

Authors’ response : The sentences from line 54 to line 59 have been moved to the Conclusion section as suggested. Please refer to lines 302-309 of the manuscript.

  1. MATERIAL AND METHODS

Line 83-84: Authors should indicate the incubation time, considering that the culture for the MIC assay must be in the semi-exponential phase.

Authors’ response : 16 h incubation period added. Please refer to line 123, 130,132

of the manuscript.

Table 1: should be removed as it does not add any additional information

Authors’ response :Table 1 removed

Line 93: Authors should be changed “doubling dilutions” in “serially two-fold dilution

Authors’ response : Changes done please refer to line 141 in manuscript.

Line 94: Mic assay is generally performed using 105 bacteria/mL (although higher concentration inoculums are used in many studies). Therefore, the Authors should indicate bacterial concentration corresponding to OD600 = 0.05.

Authors’ response : Text was originally written as “with OD600 adjusted to 0.05”. Please refer to line 143 of the manuscript.

Line 101: Are the authors evaluating AgZ for inhibition of biofilm formation (and not for degradation of a formed biofilm)? If the answer is yes, they should change "The biofilm density of AgZ treated biofilm" to "Biofilm inbition activity by AgZ"

Authors’ response : Change of wording from  "The biofilm density of AgZ treated biofilm" to "Biofilm inhibition activity by AgZ" is done as suggested. Please refer to line 135 of the manuscript.

  1. RESULTS

About MIC assay. The data shown in table 2 are partially in contrast with those of table 3. This result could depend on a different release over time of the Ag ions from the various AgZ complexes. Therefore, I suggest adding an Ag-release test over time (up to 16 hours) to validate the results obtained.

Authors’ response : Results in Table 2 show the increasing concentration of Ag-ion content in AgZ1 across to AgZ4 which has been ion exchanged in the framework of the zeolite; i.e, the Ag ion content in the zeolite will be increased in direct correlation with the molarity of the silver nitrate. Results in Table 3 shows the decrease of the MIC value with increasing Ag-ion loads in the zeolite sample (AgZ4 showing the lowest MIC, i.e lowest dilution of AgZ4 neede to kill the bacteria). The appearance of ‘contrasting results’ mentioned is a typical representation of effectiveness of Ag ion in antimicrobial activity where the higher the Ag ion is present the less concentration/dilution of the AgZ4 sample is needed to kill the bacteria. Therefore the authors are of the opinion that  an Ag-release test for this study is not needed as the results concur as in a typical MIC reaction .

About Biofilm inhibition. The Authors should also add the values of the other AgZ (AgZ1, AgZ2 and AgZ3). From the MIC data, I would expect similar data between AgZ3-AgZ4 for parahemolyticus.

Authors’ response : Due to Covid-19 lockdown related time constraints for laboratory access, the authors decided to select only the the most effective AgZ sample in MIC assay for further biofilm and SEM assays. Therefore other AgZ (AgZ1-Ag3) samples were not tested for biofilm and SEM studies.

About Figure 1. The Authors should indicate the number of replicates from which the average value was obtained. In addition, if the replicas are at least 5, a test-anova should be done to assess whether the differences between the two strains are significant.

Authors’ response : Two technical replicates were used in this study . Text is added please refer to line 126 of the manuscript.

I would also suggest doing other tests, such as live/dead or XTT, to increase biofilm inhibition results in terms of live and dead cell ratio and toxicity, respectively.

Authors’ response : Biofilm inhibition assays are in our opinion most accurately     measured by dyes that assess the extracellular matrix (ECM) content that constitute the bacterial biofilms. Crystal violet is a popular methodology used in many studies.  Live/dead or XTT biofilm assays measure proliferation ability, vitality, the metabolism of biofilms, while we only aim to evaluate physical matrix reduction and structure damage or negative effects

About Scanning Electron Microscopy (SEM). Similar to the Biofilm assay, I would suggest to increase the data with the other AgZ (AgZ1, AgZ2 and AgZ3).

Authors’ response : Due to Covid-19 lockdown related time constraints for laboratory access, the authors decided to select only the the most effective AgZ sample in MIC assay for further biofilm and SEM assays. Therefore other AgZ (AgZ1-Ag3)  samples were not tested for biofilm and SEM studies at this stage.

Please refer to updated manuscript for changes.

Reviewer 4 Report

This paper describes a study that focused on antibacterial activity and biofilm disturbance by Ag-containing zeolites. The results showed that one of the zeolites, expectedly the version that contained the highest amount of silver, had the highest anti-biofilm effect and lowest MIC. 

Although the topic is important, in my opinion the study lacks explanation of the surfaces that would be the targets for Ag-zeolite treatment. Also, the mode of application of zeolites is unclear - would these be added as a suspension, or integrated to a surface. If the latter, then the situation would be likely different from the results of this study. Also, what would be the shelf-life of the zeolites and what is the amount of Ag that migrates out from the zeolites in time? Therefore, the conclusion "zeolite A which can be applied in the aquaculture industry to combat against infectious pathogens in particular V. campbellii and V. parahaemolyticus." looks relatively premature to me.

Specific comments:

1. Introduction is extremely brief and does not introduce any current strategies to combat bacterial biofilms in aquatic systems. A couple of examples are mentioned but in my view, these are insufficient.

2. Table 1 claims to show the concentrations of zeolites tested with bacteria for MIC analysis but currently, this information is not there. I suggest removing this table.

3. The MIC concentrations of zeolites are given in Table 2 and presumably these are expressed as mg zeolite/ml water. Do the authors have data about MIC of nonAg-loaded zeolite, for reference?

4. For biofilm assay, the bacteria were incubated together with zeolites while the biofilm formation ability of the bacteria was followed. This is of course one way of analysing the anti-biofilm activity of a compound but does this reflect the actual exposure scenarios? Also, for biofilm formation, only one zeolite was selected. Did the authors consider testing any other Ag-zeolites or non Ag-loaded zeolites for reference?

4. It looks like Ag content defines the antibacterial and anti-biofilm activity of zeolites. How do the authors see the role of zeolites. 

Author Response

Comments and Suggestions for Authors (Reviewer 4)

This paper describes a study that focused on antibacterial activity and biofilm disturbance by Ag-containing zeolites. The results showed that one of the zeolites, expectedly the version that contained the highest amount of silver, had the highest anti-biofilm effect and lowest MIC. 

Although the topic is important, in my opinion the study lacks explanation of the surfaces that would be the targets for Ag-zeolite treatment. Also, the mode of application of zeolites is unclear - would these be added as a suspension, or integrated to a surface. If the latter, then the situation would be likely different from the results of this study. Also, what would be the shelf-life of the zeolites and what is the amount of Ag that migrates out from the zeolites in time? Therefore, the conclusion "zeolite A which can be applied in the aquaculture industry to combat against infectious pathogens in particular V. campbellii and V. parahaemolyticus." looks relatively premature to me.

Author’s response: The mechanism of bactericidal action of ion-exchanged zeolite has been studied with the proposed mechanism. Matsumura (2003) reported the mechanism of action in which silver ion itself released from zeolite and also the formation of reactive oxygen species from silver in the zeolites, for antibacterial effect. Overall, zeolite with ion-exchanged has been reported to be effective on the general bacteria population. The shelf-life of ion-exchanged zeolites will need to be further investigated as this has not been conducted based on this paper.

The potential application for the ion-exchanged zeolite A in the aquaculture will be its usage as a part of the ceramic filtration system either embedded in or onto a substrate can be used to inhibit the marine pathogens in killing the larvae of fish.

Specific comments:

  1. Introduction is extremely brief and does not introduce any current strategies to combat bacterial biofilms in aquatic systems. A couple of examples are mentioned but in my view, these are insufficient.

Author’s response: Extra references are added into body of introduction

  1. Table 1 claims to show the concentrations of zeolites tested with bacteria for MIC analysis but currently, this information is not there. I suggest removing this table.

Author’s response: Table 1 has been removed as suggested.

  1. The MIC concentrations of zeolites are given in Table 2 and presumably these are expressed as mg zeolite/ml water. Do the authors have data about MIC of nonAg-loaded zeolite, for reference?

Author’s response: Studies were carried out in tandem with evaluation against CuZ titled “Antibacterial Activity of Different Copper Ion-Loading Zeolite A Against Vibrio Campbellii and Vibrio Parahaemolyticus” and data was presented at the ‘International Conference on Engineering Education & Research 2019 (FICEER2019) in Saudi Arabia’. In general, CuZ (of Zeolite A) showed excellent antibacterial properties against Vibrio Campbellii and Vibrio Parahaemolyticus.

Details of the study are as in the abstract below:

ABSTRACT

Bentonite based zeolite A was hydrothermally synthesized with 30 minutes of aging followed by crystallization for 8 hours at 100oC.  In this study, the optimum silica to alumina ratio and alkalinity of alkaline activator (NaOH) in the reaction mixture were set at 1.5 and 2.5M respectively. Zeolite A obtained underwent ion-exchange process with four different concentrations of Cu(NO3)2.5H2O (0.25M, 0.50M, 1.00M and 1.5M) for 3 days. The antibacterial properties copper loaded zeolite A, (CuZ1, CuZ2, CuZ3 and CuZ4) against V. campbellii and V. parahaemolyticus were evaluated by determining the Minimum Inhibitory Concentration (MICs) using two-fold serial dilutions of ion loaded zeolites in Tryptic Soy broth (TSB) supplemented with 2% of NaCl. The concentrations CuZ ranged from 5mg/ml – 25mg/ml. After 16 h exposure of bacteria to CuZ, the antibacterial activity was determined qualitatively based on the zone of inhibition formed. The MICs value for CuZ1, CuZ2, CuZ3 and CuZ4 against V. campbellii and V. parahaemolyticus were found to be (25mg/ml, 25mg/ml, 12.5mg/ml and 6.25mg/ml) and (20mg/ml, 20mg/ml, 10mg/ml and 5mg/ml) respectively. It indicates that V. campbellii are more resistant compared to V. parahaemolyticus. The results obtained prove that CuZ show remarkable antibacterial properties against V. campbellii and V. parahaemolyticus.

Authors’ response: 2) The study employed negative controls throughout the experiment which comprised media and non-Ag zeolite added with bacteria. End result of negative control showed full bacterial growth across the media.

  1. For biofilm assay, the bacteria were incubated together with zeolites while the biofilm formation ability of the bacteria was followed. This is of course one way of analysing the anti-biofilm activity of a compound but does this reflect the actual exposure scenarios? Also, for biofilm formation, only one zeolite was selected. Did the authors consider testing any other Ag-zeolites or non Ag-loaded zeolites for reference?

Authors’ response: 1) Crystal Violet assay for biofilm formation evaluation is one of the most popular quantitative methods employed for biofilm determination, apart from CFU/ml Viable Growth determination. The assay evaluates the effect of zeolites on the amount extracellular matrix (ECM) of the biofilms via absorbance readings of the crystal violet at 570nm on a UVVIS spectrophotometer. This assay is suitable to the authors’ opinion as we are looking into the detrimental effect of zeolite on the ECM of the biofilms. This observation was then concurred by Scanning Electron Microscopy which showed obvious structural damages to the ECM network. 

2) The study employed negative controls throughout the experiment which comprised media and non-Ag zeolite added with bacteria. End result of negative control showed full bacterial growth across the media. The images of MIC plates are available upon request

  1. It looks like Ag content defines the antibacterial and anti-biofilm activity of zeolites. How do the authors see the role of zeolites. 

Authors’ response: Due to its properties of being non-toxic, having excellent durability and heat resistance, zeolite have been used as inorganic reservoirs for metal ions (İyİgündoğdu et al., 2014). In general, zeolite has the ability to regulate the release of metal ion. Regulating the release of metal ion will provide a sustained or prolong the antibacterial action of the material (Zhao et al., 2006; Kwakye-Awuah et al., 2008).

References:

1.KwakyeAwuah, B., Williams, C., Kenward, M. A., & Radecka, I. (2008). Antimicrobial action and efficiency of silverloaded zeolite X. Journal of Applied Microbiology, 104(5), 1516-1524.

2.İYİGÜNDOĞDU, Z., Demirci, S., Bac, N., & ŞAHİN, F. (2014). Development of durable antimicrobial surfaces containing silver-and zinc-ion-exchanged zeolites. Turkish Journal of Biology, 38(3), 420-427.

3. Zhao, R., Guo, F., Hu, Y., & Zhao, H. (2006). Self-assembly synthesis of organized mesoporous alumina by precipitation method in aqueous solution. Microporous and mesoporous materials, 93(1-3), 212-216.

Please refer to attached manuscript for latest edits. Thanks

Round 2

Reviewer 1 Report

In this form the paper can be accepted for publication in Applied Sciences.

Author Response

Thank you for your favourable comment.

Reviewer 3 Report

Authors have partially been replied at the comments, also due to difficulty related to Coronavirus pandemic. However, I have many doubts about MIC values (that should be coupled with Ag-release test over time from AgZ complexes) and Biofilm data (missing of the other AgZ complexes, namely AgZ1, AgZ2 and AgZ3).

Therefore, I am sorry to have to suggest reject of the manuscript.

Author Response

The authors would like to thank the reviewer for reviewing our manuscript.

Reviewer 4 Report

The authors have answered several of my concerns. However, there are still some remaining issues they did not address properly.

  1. I was asking about the negative controls, i.e., bacterial/biofilm exposure to non Ag-laden zeolites but zeolite A without any Ag (or other potentially antibacterial element) added. Currently, as I understand, the control is non zeolite treated biofilm/bacterial suspension. It is possible that zeolites by themselves have an effect on bacterial cells. Have the authors tested that?
  2. I would have liked the authors to show how much Ag is released from zeolites, i.e., what is the bacteria-effective concentration. This question relates also with the reusability of the zeolites. i.e., how long can one zeolite particle be used before it leaches all the Ag it was impregnated with. Although the authors claim this will be studied in their next article, it would have been informative to know at least about the release of Ag from zeolites.
  3. The authors have erroneously stated that crystal violet assay was used to quantify biofilm activity. Please correct this.

Author Response

Author's response to comments:

  1. I was asking about the negative controls, i.e., bacterial/biofilm exposure to non Ag-laden zeolites but zeolite A without any Ag (or other potentially antibacterial element) added. Currently, as I understand, the control is non zeolite treated biofilm/bacterial suspension. It is possible that zeolites by themselves have an effect on bacterial cells. Have the authors tested that?Response: File of zeolite only control is attached as "AgZ Controls MIC." File will show images of zeolite only sample (no Ag loads) giving high numbers of bacterial group.
  2. Error on ' activity' changed to 'growth'. Please refer to manuscript Lines 135- 136. 
  3. I would have liked the authors to show how much Ag is released from zeolites, i.e., what is the bacteria-effective concentration. This question relates also with the reusability of the zeolites. i.e., how long can one zeolite particle be used before it leaches all the Ag it was impregnated with. Although the authors claim this will be studied in their next article, it would have been informative to know at least about the release of Ag from zeolites.Authors response : Ag release will be conducted in next level of study.

Round 3

Reviewer 3 Report

I suggest the publication.